# Arginine Regulates Skeletal Muscle Fiber Type Formation via mTOR Signaling Pathway

**DOI:** 10.3390/ijms25116184

**Published:** 2024-06-04

**Authors:** Min Zhou, Yihan Wei, Yue Feng, Shumin Zhang, Ning Ma, Kaige Wang, Peng Tan, Ying Zhao, Jinbiao Zhao, Xi Ma

**Affiliations:** State Key Laboratory of Animal Nutrition and Feeding, College of Animal Science and Technology, China Agricultural University, Beijing 100193, China; 18854811609@163.com (M.Z.); www116623@163.com (Y.W.); yuefeng@cau.edu.cn (Y.F.); zhangshumin@mail.tsinghua.edu.cn (S.Z.); maningrebecca@cau.edu.cn (N.M.); wkg960913@126.com (K.W.); tanpeng1995@cau.edu.cn (P.T.); yingzhaocaas@163.com (Y.Z.); jinbiaozhao@cau.edu.cn (J.Z.)

**Keywords:** arginine, skeletal muscle, fiber type, mitochondrial biogenesis, mTOR signaling pathway

## Abstract

The composition of skeletal muscle fiber types affects the quality of livestock meat and human athletic performance and health. L-arginine (Arg), a semi-essential amino acid, has been observed to promote the formation of slow-twitch muscle fibers in animal models. However, the precise molecular mechanisms are still unclear. This study investigates the role of Arg in skeletal muscle fiber composition and mitochondrial function through the mTOR signaling pathway. In vivo, 4-week C56BL/6J male mice were divided into three treatment groups and fed a basal diet supplemented with different concentrations of Arg in their drinking water. The trial lasted 7 weeks. The results show that Arg supplementation significantly improved endurance exercise performance, along with increased SDH enzyme activity and upregulated expression of the MyHC I, MyHC IIA, PGC-1α, and NRF1 genes in the gastrocnemius (GAS) and quadriceps (QUA) muscles compared to the control group. In addition, Arg activated the mTOR signaling pathway in the skeletal muscle of mice. In vitro experiments using cultured C2C12 myotubes demonstrated that Arg elevated the expression of slow-fiber genes (MyHC I and Tnnt1) as well as mitochondrial genes (PGC-1α, TFAM, MEF2C, and NRF1), whereas the effects of Arg were inhibited by the mTOR inhibitor rapamycin. In conclusion, these findings suggest that Arg modulates skeletal muscle fiber type towards slow-twitch fibers and enhances mitochondrial functions by upregulating gene expression through the mTOR signaling pathway.

## 1. Introduction

Skeletal muscle comprises approximately 40% of an individual’s body mass [1], and is the most abundant tissue of mammals. The proportion of muscle is closely related to its critical functions. It not only affects human exercise ability and health [2], but also directly impacts the yield and quality of meat in livestock and poultry [3]. Skeletal muscle is generally classified into fast- and slow-twitch fiber types based on differences in oxidative capacity, myoglobin content, and mitochondrial levels [4]. Currently, the most common classification method involves dividing the myosin heavy chain (MyHC) into four types: slow oxidation type (MyHC I), fast oxidation type (MyHC IIA), intermediate type (MyHC IIX), and fast glycolytic type (MyHC IIB) [5]. In animal husbandry, muscle fiber types are closely associated with meat pH, color, drip loss, and tenderness [6]. Increasing the proportion of MyHC I is favorable for improving meat quality [7]. In humans, slow and fast muscle fibers have a direct impact on endurance and explosive power, respectively. Specifically, increasing the proportion of MyHC I can improve the physical fitness of resistance training athletes, such as weightlifters. Conversely, increasing the proportion of MyHC IIA can enhance the performance of athletes who rely on explosive power, such as sprinters [8]. Therefore, regulating skeletal muscle fiber composition plays a crucial role in enhancing meat quality and exercise capacity.

It is important to note that the composition of muscle fibers is not constant, and it can be influenced by factors such as animal age, gender, hormones, external nutrition, exercise, and the environment, which can affect the transformation between fiber types. Among these factors, dietary nutrients play a crucial role in skeletal muscle development both before and after birth. Nutrients such as leucine [9], L-arginine [10], succinate [11], and naringin [12] are known to be important for muscle fiber-type remodeling. L-Arginine (Arg) is a semi-essential amino acid (AA) for animals. When the body is in a state of hunger, injury, stress, or rapid growth, endogenous Arg synthesis may not meet the body’s requirements. Hence, it is necessary to supply exogenous Arg. Arg exerts various physiological functions in animals, including mitochondrial function, fat deposition, reproductive performance, and antioxidant activity [13,14,15]. Recent studies have reported that Arg promotes the formation of type I muscle fibers in mice, weaning piglets, and porcine skeletal muscle satellite cells. The underlying mechanisms involve the Sirt1/AMPK, Akirin2, and AMPK/PGC-1α signaling pathways [10,16].

As a sensor of nutrition, mTOR plays a role in regulating lipid and protein metabolism, cell survival, cell death, and aging [17,18,19]. Additionally, mTOR modulates mitochondrial biogenesis and turnover [20,21]. Furthermore, mitochondrial function is closely linked to muscle fiber type [22]. However, it remains unclear whether Arg affects muscle function, muscle fiber formation, and mitochondrial biogenesis through the mTOR signaling pathway. Therefore, we hypothesize that Arg can regulate gene expression associated with fiber type in skeletal muscle via the mTOR pathway.

To test this hypothesis, we first examined the effects of Arg on exercise tolerance, energy metabolism enzymes, and skeletal muscle fiber composition. By using both in vivo and in vitro models, we demonstrated that Arg increases the proportion of slow-twitch skeletal muscle through the activation of the mTOR signaling pathway. Additionally, we investigated the effects of rapamycin on genes related to muscle fiber type and mitochondrial function in C2C12 cells. In this article, we present the initial evidence suggesting that the mTOR signaling pathway may serve as a novel mechanism through which Arg regulates muscle fiber type and mitochondrial function.

## 2. Results

### 2.1. Growth Performance and Organ Index

To determine whether Arg has effects on skeletal muscle growth, we firstly fed C57BL/6J male mice with a chow diet supplemented with 0, 0.5%, or 1.0% Arg for 7 weeks, which refers to the dosages selected in other studies [10,16]. We recorded body weights and food intake of mice every week during the 7 weeks of treatment. The results showed no significant difference in the initial and final body weight of the mice (Appendix A). We also found that Arg supplementation had no effects on the body weight gain (Appendix A), food intake (Appendix A), and organ index, including tibialis anterior muscle (TA) (Appendix A), soleus muscle (SOL) (Appendix A), quadriceps femoris (QUA) (Figure 1A), gastrocnemius muscle (GAS) (Figure 1B), brown adipose tissue (BAT), epididymal white adipose tissue (eWAT), liver, testis, and kidney (Appendix A–L), while the inguinal white adipose tissue (iWAT) index showed a downward trend (Appendix A).

### 2.2. Exogenous Arg Improves Endurance Exercise Ability and Shifts Skeletal Muscle Fiber Size Distribution

To further explore the role of Arg on skeletal muscle contraction properties, we tested the exercise capacity of the mice and observed that Arg increased low-speed running time (*p* < 0.05, Figure 1C). However, supplementing Arg did not change high-speed running time (Figure 1D), suggesting Arg improves endurance exercise performance, instead of explosive exercise performance. Interestingly, we also found that the 0.5% and 1.0% Arg augmented the number of muscle fibers per unit area in QUA (*p* < 0.05, Figure 1E,G), GAS (*p* = 0.089, Figure 1F,I), and small muscle fiber proportion (1000–2000 μm^2^). Arg decreased large muscle fiber proportion (2000–4000 μm^2^; Figure 1H,J). In addition, exogenous Arg did not influence the histomorphology and the number of muscle fibers per unit area of the TA (Appendix A) and SOL muscles (Appendix A). The alternation of muscle fiber size distribution indicates that Arg has an impact on skeletal muscle contraction properties.

Next, we further explored the metabolism enzyme activity in serum, liver, and muscle. We observed that Arg suppressed the activity of lactic dehydrogenase (LDH; Figure 1K) in serum, and increased succinate dehydrogenase activity (SDH; Figure 1L,N) in serum and liver. Both in GAS and QUA, LDH enzyme activity showed a downward trend in the Arg-supplemented treatment (Figure 1O,Q). Moreover, Arg had no obvious effects on the activity of LDH in liver, and SDH in QUA and GAS (Figure 1M,P,R).

### 2.3. Arg Increases Proportion of Slow-Twitch Fibers In Vivo

MyHC I and MyHC IIA are the major muscle fiber types that affect endurance exercise performance [23]. Based on our above finding that Arg can improve muscle endurance in mice, we further explored whether Arg affects muscle fiber-type transformation in GAS (fast- and slow-twitch fibers) and QUA (fast-twitch muscle), focusing on changes in MyHC I and MyHC IIA, which are related to endurance performance. Each skeletal muscle fiber, containing I, IIA, IIX, and IIB, expresses diverse myosin heavy chain and troponin isoforms. Immunofluorescence staining of slow muscle fibers MyHC I (green) and fast muscle fibers MyHC IIX (red) showed that Arg doubled the percentage of slow-twitch fiber MyHC I in QUA and GAS (Figure 2A–D). Furthermore, 0.5% and 1.0% Arg increased MyHC I, MyHC IIX, and Tnnt1 in fast QUA muscle, and 1.0% Arg upregulated *MyHC IIA* and deceased fast-twitch fiber-related gene *MyHC IIB* mRNA levels (Figure 2E). Consistently, in the mixed GAS muscle, Arg promoted the relative mRNA expression of *MyHC I*, *MyHC IIA*, *MyHC IIX*, and *Tnnt1* (Figure 2F). Additionally, Arg downregulated *MyHC IIB* mRNA levels in GAS (Figure 2F). Arg promoted the protein expression of MyHC I and MyHC IIA in QUA (Figure 2G) and tended to increase the expression of the MyHC I and MyHC IIA proteins in GAS, but there was no statistical difference (Figure 2H). In summary, Arg promotes the transformation of slow-twitch fibers and has the function of improving muscle endurance.

### 2.4. Arg Promotes Mitochondrial Biogenesis in Skeletal Muscle

The above results indicate that Arg may enhance aerobic metabolism. To further support this hypothesis, an increased mitochondrial biogenesis was consistently observed in this study. We detected mitochondrial biogenesis-associated genes, such as *PGC-1α*, *MEF2C*, *TFAM* and *NRF1* (Figure 3A,B). We also checked the protein expression of PGC-1α (Figure 3C,D). The results show that the mRNA level of *PGC-1α* and *NRF1* were dose-dependently increased by Arg in QUA (Figure 3A). An amount of 0.5% Arg significantly promoted the PGC-1α protein expression in QUA (Figure 3C). In the mixed GAS, the addition of 0.5% arginine from an external source resulted in the upregulation of mRNA levels for *PGC-1α*, *MEF2C*, *TFAM*, and *NRF1* (Figure 3B). While there was an observed enhancement in PGC-1α protein levels, this did not reach statistical significance, possibly due to variations within the group (Figure 3D).

### 2.5. Arg Activates mTOR Signaling Pathway in Skeletal Muscle of Mice

To further explore the mechanism of Arg regulating muscle fiber types, we analyzed two signaling pathways closely related to this study. The results showed that there was no remarkable effect of arginine on p-AMPK protein levels in QUA and GAS muscles (Figure 4A,B). Although the phosphorylation level of AMPK reduced at 0.5% Arg, it was not statistically different (Figure 4A,B). The protein level of p-AKT showed an upward trend in QUA and GAS (Figure 4C,D). Meanwhile, we found that 1.0% Arg could enhance the phosphorylation level of mTOR in QUA and GAS (Figure 4E,F).

### 2.6. Arg Affects Fiber Type and Mitochondrial Function via mTOR Signaling Pathway in C2C12 Myotubes

To test the role of Arg in regulating muscle fiber type, we used C2C12 cells as an in vitro model. The concentration of 0.4 mM arginine in the control group was the concentration of arginine contained in the growth medium (high-glucose DMEM medium) without additional arginine. The concentrations of 1.2 mM and 3.6 mM arginine in the treatment group were in reference to the dosages of arginine treated with the C2C12 cell line in other studies [16,24,25]. Consistent with cell morphology (Figure 5A), Arg significantly promoted the mRNA expression of myomarker (Figure 5D), suggesting Arg can promote myoblast myotube fusion. Moreover, we tested the enzyme activity of LDH and SDH (Figure 5B,C); meanwhile, in this study, 1.2 mM of Arg did not significantly affect the above indicators, which may be related to Arg concentration and intervention time. Similar to the in vivo results, Arg increased the mRNA expression of *MyHC I*, *PGC-1α*, *TFAM*, *MEF2C*, and *NRF1*, while decreasing *MyHC IIB* mRNA levels (Figure 5E,F). Moreover, Arg promoted the protein levels of PGC-1α, p-mTOR, and p-P70S6K (Figure 5G,H).

Furthermore, rapamycin significantly inhibited the expression of p-mTOR during C2C12 myogenic differentiation (Figure 6G). The mRNA levels of *MyHC I*, *MyoG*, *Myomarker*, *PGC-1α*, *Tnnt1*, and *MEF2C* were significantly reduced in rapamycin treatment for 4 (Figure 6A–C) or 6 days (Figure 6D–F). However, rapamycin did not influence the mRNA level of *NRF1* (Figure 6C,F). Similarly, the protein expression of MyHC I was decreased in the rapamycin group after 4 or 6 days (Figure 6H).

## 3. Discussion

Arg has been extensively studied as a dietary supplement for its potential effects on various biological functions, including antioxidant properties [14], anti-inflammatory properties [26], regulation of lipid metabolism [15,27,28], mitochondrial function [10], and muscle fiber type transformation [11,16]. These effects are relevant to improving human mobility and the quality of livestock and poultry meat. Arg is classified as a semi-essential or conditionally essential amino acid in mammals of different ages and physiological states. In our study, we found that Arg did not significantly affect feed intake or weight gain in mice. This finding is consistent with previous reports that Arg did not impact average daily weight gain, food intake, or feed/gain ratio in finishing pigs [29]. However, other studies have shown that Arg can enhance the growth performance of broiler chickens challenged with coccidiosis [26]. It is important to note that the optimal dietary dosage of Arg may vary among different animal species, ages, physiological states, and intervention times. For instance, Chen et al. found that 1.0% Arg supplementation may be optimal for weaning piglets [10]. Overall, the differences in the effects of Arg on feed intake and body weight gain can be attributed to the specific characteristics of the experimental animals, dosage, and intervention timing.

Muscle fiber types are categorized into different types based on their expression of specific myosin heavy chain (MyHC) isoforms, including type I (MyHC I) for slow oxidative fibers, type IIA (MyHC IIA) for fast oxidative fibers, type IIX (MyHC IIX) for intermediate oxidative and glycolytic fibers, and type IIB (MyHC IIB) for fast glycolytic fibers [30]. It is widely accepted that the number of muscle fibers is determined before birth [31], while the regulation of muscle fibers after birth primarily relies on changes in morphology (diameter, length, etc.) and muscle fiber type. Muscle fiber type is closely associated with meat quality indicators such as pH value, water-holding capacity, meat color, and tenderness [32]. Therefore, gaining a deep understanding of the regulatory mechanisms underlying muscle fiber types is crucial for improving meat quality. In this study, we observed that exogenous Arg upregulated the expression of genes associated with slow muscle fibers both in mouse muscle tissue and C2C12 cells. A previous study in Kunming mice demonstrated that dietary Arg supplementation increased slow MyHC expression and decreased fast MyHC expression in the tibialis anterior muscle [16]. Additionally, our findings revealed that the effect of Arg on muscle fiber types was not dose-dependent. Specifically, 1.0% Arg supplementation decreased MyHC I expression and increased MyHC IIB expression compared to 0.5% Arg supplementation [16]. Similarly, in the gastrocnemius muscle, we observed a reduction in MyHC I mRNA expression with 1.0% Arg supplementation. These results suggest that the dosage of Arg, intervention time, muscle fiber type composition (e.g., tibialis anterior, soleus, gastrocnemius, and quadriceps), and animal species can collectively influence the impact of Arg on muscle fiber types. In addition, it is important to acknowledge certain limitations in investigating the effects of arginine on different muscle fiber types. Specifically, due to the small size and close proximity of various types of skeletal muscles, there is a possibility of inadvertently collecting nearby muscles during sampling, which could potentially influence the analysis of the collected samples. For instance, the gastrocnemius and plantaris muscles have a close distribution, and it is possible that some plantaris muscle samples were inadvertently collected along with the gastrocnemius samples, potentially affecting the obtained results.

In pig models, the supplementation of 1.0% Arg was found to upregulate the expression of genes related to slow muscle fibers (MyHC I, Tnnt1, Tnnc1, and Tnni1) in the longissimus dorsi muscle [10]. Additionally, the composition of muscle fiber types is closely associated with metabolic enzymes, and the metabolic characteristics of muscle fibers can influence the activities of these enzymes [33]. Previous research has indicated that slow muscle fibers exhibit higher succinate dehydrogenase (SDH) enzyme activity, while fast muscle fibers exhibit higher lactate dehydrogenase (LDH) enzyme activity [34]. The activities of muscle fiber-related metabolic enzymes not only affect exercise performance but also impact energy metabolism in postmortem muscles, including glycogen levels, lactate content, and pH. Numerous studies have shown that Arg supplementation increases SDH and malate dehydrogenase (MDH) activities while decreasing LDH activity in skeletal muscle and C2C12 myotubes [10,16]. Consistent with these findings, our study also found that Arg enhanced SDH activity and suppressed LDH activity, which corresponded to increased endurance and a higher proportion of oxidative muscle fibers in mice.

Mitochondria play a crucial role in providing energy for eukaryotic cells [35]. Generally, the oxidative capacity of tissues or cells is enhanced by increasing the number of mitochondria. Mitochondrial biogenesis is regulated by both mitochondrial and nuclear genes. The activity of these genes is controlled by transcription factors such as NRF1, TFAM, MEF2C, Sirt1, and PGC-1α [36]. PGC-1α, a key regulator of oxidative metabolic enzymes and mitochondrial biosynthesis, is expressed in skeletal muscle [36]. PGC-1α regulates the transcriptional activity of TFAM [37], which promotes mitochondrial DNA (mtDNA) replication and transcription. The interaction between TFAM and mtDNA is involved in mitochondrial biogenesis [38]. PGC-1α activates downstream nuclear transcription factors, including NRF1, to regulate mitochondrial biogenesis [39]. Overexpression of PGC-1α can enhance anti-fatigue ability by increasing mitochondrial content and oxidative enzyme levels [40]. Furthermore, promoting mitochondrial biogenesis is crucial for facilitating the formation of MyHC I and can serve as a novel mediator for muscle fiber type transformation [41]. Nutrients such as resveratrol, leucine, and succinate can influence the expression of slow muscle fibers by improving mitochondrial biogenesis [42,43]. A recent study demonstrated that 1.0% Arg promoted the protein expression of PGC-1α, Sirt1, and cytochrome C, as well as the mRNA expression of PGC-1α, NRF1, TFB1M, and ATP5G in muscle, indicating that Arg plays a significant role in regulating mitochondrial biogenesis [10]. Consistent with these findings, our study revealed that Arg upregulated the expression of PGC-1α, TFAM, and NRF1 in muscles, further supporting the role of Arg in improving mitochondrial biogenesis through the regulation of these aforementioned genes.

The AMPK and mTOR signaling pathways are crucial energy receptors involved in energy metabolism and biological activities [44,45]. In this study, the p-AMPK/AMPK ratio did not show significant changes in the GAS and QUA muscles, suggesting that the AMPK signaling pathway might not play a prominent role in this particular study. This observation could be attributed to factors such as the timing and dosage of Arg intervention, as well as the specific muscle fiber types under investigation. Additionally, previous research conducted in our laboratory demonstrated that increasing extracellular Arg concentration activates the mTOR signaling pathway in brown adipocyte precursor cells, promoting their growth and development in a dose-dependent manner [45]. Another interesting study revealed that the L-Arg/NO/mTOR/p70S6K signaling pathway promotes muscle development in fast-twitch fibers rather than slow-twitch fibers in chickens [46].

While there is no direct evidence indicating that the mTOR signaling pathway can directly regulate muscle fiber types, it has been shown that mTORC1 can influence the expression and function of PGC1, thereby impacting mitochondrial function [47,48]. The Akt-mTOR signaling pathway is a crucial upstream pathway involved in mitochondrial metabolism [49]. Here, we observed an increase in AKT phosphorylation in the GAS muscle following Arg supplementation. Phosphorylation of Akt facilitates mTOR activation, which is essential for mitochondrial metabolism. Enhanced activation of mTOR has been shown to significantly promote mitochondrial biogenesis in satellite cells [49]. Previous studies have indicated that mitochondrial function declines differently in fast-twitch and slow-twitch muscles with age, suggesting that mitochondrial dynamics vary according to the fiber type [50]. MyHC I and IIA fibers exhibit a higher rate of mitochondrial fusion compared to MyHC IIX/IIB fibers [50]. Furthermore, fused mitochondrial networks have been associated with improved metabolic states and may contribute to ATP homeostasis in oxidative fibers [51]. Additionally, studies have shown that L-leucine activates the mTOR signaling pathway to regulate muscle fiber type [52], suggesting the significance of mTOR in muscle fiber type determination. It has been reported that type II fibers have a higher capacity for the mTOR pathway [53]. Consistent with these findings, our research showed higher levels of mTOR phosphorylation in the QUA muscle compared to the GAS muscle. Based on this, we speculate that the mTOR signaling pathway serves as a key mediator of Arg-induced muscle fiber remodeling. Supporting this hypothesis, we observed that rapamycin, an mTOR inhibitor, suppressed the expression of genes related to mitochondrial biogenesis and the formation of slow myosin fibers. Further research is needed to elucidate the precise mechanisms underlying these effects and to determine the optimal dosage and timing of Arg supplementation for different animal species and physiological conditions.

## 4. Materials and Methods

### 4.1. Animal Experiments

C57BL/6J mice (Issue No. AW60702202-1-5, Beijing HFK Bioscience Co., Ltd., Beijing, China) were housed under controlled conditions of temperature (23 ± 3 °C) and relative humidity (60 ± 10%) with a 12 h light/12 h dark cycle. The mice (n = 6) were randomly divided into three groups based on their body weight. The groups received standard diets with drinking water containing 0%, 0.5%, or 1.0% Arginine (Arg) (Sigma-Aldrich, Macquarie Park, NSW, Australia). The dosage of arginine was in reference to other studies [10,16]. The bodyweight and food intake of the mice were recorded weekly. Low-speed running tests were performed in the fifth week, followed by fast-speed running tests in the sixth week. After 7 weeks, the mice were euthanized using diethyl ether anesthesia, and samples of blood, serum, skeletal muscles, kidney, fat, liver, and testis were collected.

### 4.2. Strength and Exercise Endurance

Strength and exercise endurance tests were conducted using the SA101B Animal treadmill (Sansbio, Nanjing, China), following a previously described protocol [54]. The mice underwent a treadmill running test with an initial speed of 10 m/min for 10 min to acclimate them. Subsequently, the mice were subjected to high-speed and low-speed running tests. The velocity for the high-speed running test was increased by 5 m/min every 2 min until reaching 40 m/min. The velocity for the low-speed running test was increased by 1 m/min every 3 min.

### 4.3. Cell Culture

The mice myoblast cell line C2C12 was cultured in high-glucose DMEM (HyClone, Logan, UT, USA) supplemented with 10% fetal bovine serum (FBS) (Gibco, Carlsbad, CA, USA), 100 μg/mL of streptomycin, and 100 U/mL of penicillin. When the cells reached 90% confluency, DMEM with 2% horse serum (HyClone, Logan, UT, USA) was added for 6 days to induce myoblast differentiation into myotubes. On day 1 of differentiation, treating C2C12 with 1.2 mM or 3.6 mM of arginine for 72 h. The dosage of arginine was in reference to other studies [16,24,25]. On days 0, 2, and 4 of differentiation, DMSO or 1 μM of rapamycin (Beyotime, Shanghai, China) was added to the complete medium to culture the cells. After 6 days of differentiation, the cells were collected to measure the relevant indicators.

### 4.4. Enzyme Activities Assay

The activity of lactic dehydrogenase (LDH) and succinodehydrogenase (SDH) was measured using commercially available assay kits (Nanjing Jiancheng Bioengineering Institute, Nanjing, China).

### 4.5. RNA Extraction, Reverse Transcript, and qRT-PCR

Total RNA was extracted from muscles and C2C12 cell lines using the Trizol reagent (Invitrogen, Carlsbad, CA, USA). Reverse transcription of RNA into cDNA was performed using a Reverse Transcriptase kit (Mei5bio, Beijing, China). Subsequently, SYBR Green PCR Supermix (Mei5bio, Beijing, China) was used to perform qRT-PCR reactions with specific primers (Sangon Biotech, Shanghai, China) according to the manufacturer’s instructions. The qRT-PCR reactions were carried out in a Real-Time PCR System LightCycler96 (Roche, Basel, Switzerland). The primer sequences used are listed in Table 1.

### 4.6. Western Blot Assay

Muscles or C2C12 cells were lysed using RIPA lysis buffer containing 1 mM of PMSF (Phenylmethanesulfonyl fluoride) and phosphatase inhibitors. Protein concentration was determined using a BCA protein assay kit (23225, Thermo Fisher Scientific Inc., Walthm, MA, USA). SDS-PAGE electrophoresis was performed, followed by incubation with primary antibodies: rabbit anti-GAPDH (K106389P, Solarbio, Beijing, China, 1:2000), rabbit anti-β-Actin (K101527P, Solarbio, 1:1000), mouse anti-MyHC I (ab11083, Abcam, Cambridge, England, United Kingdom, 1:1000), mouse anti-MyHC IIa (11128-1-AP, Proteintech, Wuhan, China, 1:1000), mouse anti-PGC-1α (66369-1-Ig, Proteintech, 1:1000), rabbit anti-AMPKα (D5A2) (#5831s, Cell Signaling Technology, Danvers, MA, USA, 1:11,000), rabbit anti-P-AMPK (Thr172) (#2535s, Cell Signaling Technology, 1:1000), rabbit anti-mTOR (#2972s, Cell Signaling Technology, 1:1000), rabbit anti-P-mTOR (#5536s, Cell Signaling Technology, 1:1000), rabbit anti-AKT (#9272s, Cell Signaling Technology, 1:1000), rabbit anti-PAKT (#9271s, Cell Signaling Technology, 1:1000), rabbit anti-p70 S6 Kinase (#9202s, Cell Signaling Technology, 1:1000), and rabbit anti-phospho-p70 S6 Kinase (Thr389) (#9205s, Cell Signaling Technology, 1:1000). The protein expression levels were analyzed using ImageJ software version 1.53.

### 4.7. Histological Analysis

For the staining of the muscle sections, we collected muscle samples in a 4% paraformaldehyde solution. The samples were dehydrated with xylene and ethanol, then embedded in paraffin. The embedded blocks were sectioned into 6 μm. After the sections were dewaxed, the sections were rehydrated with xylene and graded concentrations of alcohol, then stained with hematoxylin solution (Solarbio, Beijing, China) for 5 min, then rinsed with water until the color can no longer be washed off. The sections were then differentiated and were immersed in water until the sections’ color turned blue; finally, they were incubated in eosin solution (Solarbio, Beijing, China) for 1 min, sealing the dehydrated tissue sections with a clear resin. The numbers and area of muscle fibers were measured using ImageJ software version 1.53.

### 4.8. Immunofluorescence Staining

Firstly, the sections were blocked with 5% BSA for 30 min and incubated overnight with the primary antibody, mouse anti-slow skeletal muscle myosin heavy chain rabbit pAb (Servicebio, Wuhan, China, GB111857, China). Subsequently, the sections were incubated with goat anti-rabbit IgG (Servicebio, GB25303, China) and FITC-TSA solution (Servicebio, G1222, China) in the dark. They were then incubated with secondary anti-fast skeletal muscle myosin heavy chain rabbit pAb (Servicebio, GB112130, China), secondary antibody (Servicebio, GB23303, China), and CY3-TSA solution (Servicebio, G1223, China) sequentially. Finally, the sections were incubated with quenching reagent (Servicebio, G1401, China) and DAPI counterstain in the dark. Images were captured using Fluorescent Microscopy (Nikon, Tokyo, Japan) and analyzed with ImageJ software version 1.53.

### 4.9. Data and Statistical Analysis

Each experiment included a minimum of three biological replicates. The data are presented as means ± SEM. Using GraphPad Prism 9.0 to carry out statistical analysis, one-way analysis of variance (ANOVA) tests was used to determine differences between the control and dose-effect groups. *p* < 0.05 was considered statistically significant.

## 5. Conclusions

In conclusion, Arg supplementation in water improved the endurance exercise performance by enhancing SDH enzyme activity and upregulating MyHC I/IIA mRNA or protein expression in the GAS or QUA muscles. These findings indicate that Arg induces a transition from fast-twitch to slow-twitch muscle fibers. Additionally, Arg supplementation promotes mitochondrial biogenesis and activates the Akt/mTOR signaling pathway, which may play a role in muscle function and muscle fiber remodeling. In vitro experiments showed that rapamycin inhibits the expression of MyHC I and genes related to mitochondrial function (Figure 7). Arg regulates skeletal muscle fiber type through mTOR signaling pathways in C2C12 cells. This research not only provides a theoretical basis for using Arg as a functional substance to regulate meat quality and enhance human exercise capacity but also presents a new mechanism by which Arg triggers an mTOR-mediated transformation from fast- to slow-twitch fiber types in skeletal muscle.

## Figures and Tables

**Figure 1 ijms-25-06184-f001:**
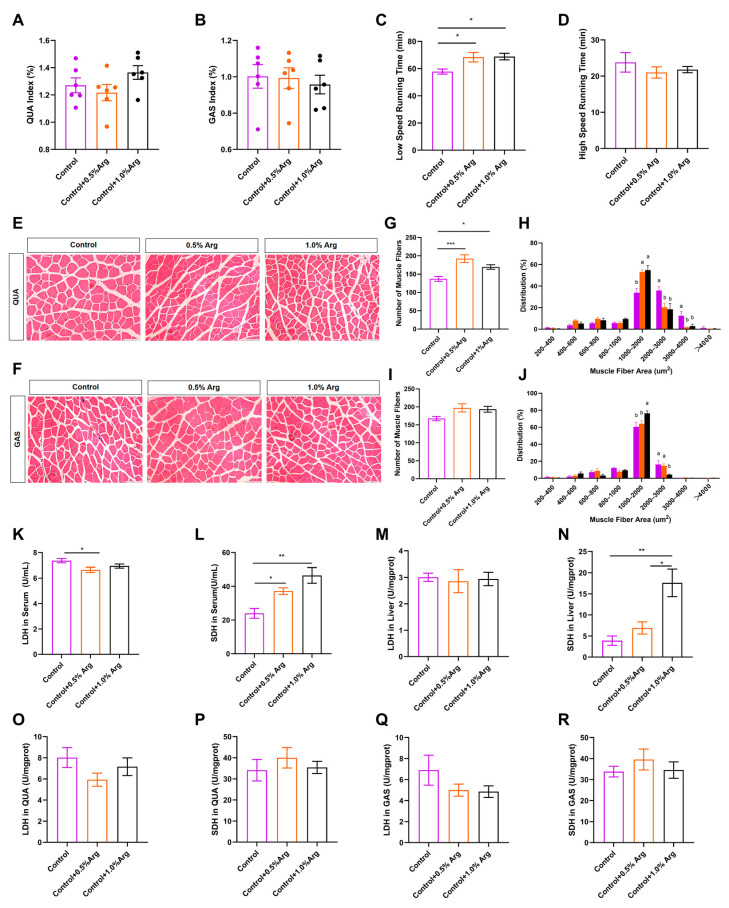
Effects of Arg on muscle histomorphology and endurance exercise capacity. Male C57BL/6J mice were fed with a chow diet supplement with 0, 0.5, and 1.0% Arg in water for 7 weeks. (**A**–**D**) Quadriceps femoris index (**A**), gastrocnemius index (**B**), low-speed running time (**C**), high-speed running time (**D**). (**E**–**J**) Representative images of HE staining of quadriceps femoris (**E**) and gastrocnemius (**F**) muscles; number of muscle fibers in a fixed area was based on HE staining in quadriceps femoris (**G**) and gastrocnemius (**I**); frequency distribution of fiber cross-sectional area of two types of muscle fibers in quadriceps femoris (**H**) and gastrocnemius (**J**). Scale bars in (**E**,**F**) represent 100 μm. (**K**–**R**) The enzyme activity of LDH in serum (**K**), liver (**M**), quadriceps femoris (**O**), and gastrocnemius (**Q**), and SDH in serum (**L**), liver (**N**), gastrocnemius (**P**), and quadriceps femoris (**R**). The purple column represents the control group with no added Arg, the orange column represents the treatment group with 0.5% Arg added. The black column represents the treatment group with 1.0% of Arg added. Means without a common letter differ, *p* < 0.05. *: *p* < 0.05, **: *p* < 0.01, ***: *p* < 0.001. Values represent the mean ± SEM.

**Figure 2 ijms-25-06184-f002:**
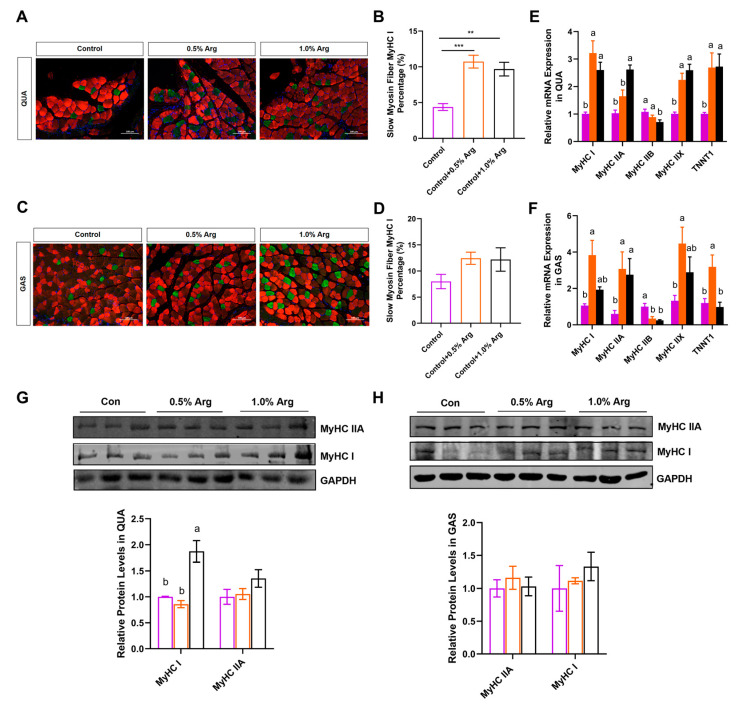
Arg promotes the expression of slow myosin fiber in mice. (**A**–**D**) Slow fiber MyHC I immunofluorescent staining (green), fast fiber MyHC IIX immunofluorescent staining (red) in quadriceps femoris (**A**), and gastrocnemius (**C**). Scale bars in (**A**,**C**) represent 100 μm. Percentage of slow fiber in quadriceps femoris (**B**), and gastrocnemius (**D**). (**E**,**F**) The relative mRNA expression of *MyHC I*, *MyHC IIA*, *MyHC IIB*, *MyHC IIX*, and *TNNT1* in quadriceps femoris (**E**), and gastrocnemius (**F**). The purple column represents the control group with no added Arg, the orange column represents the treatment group with 0.5% Arg added. The black column represents the treatment group with 1.0% of Arg added. (**G**,**H**) Immunoblots and quantification of MyHC I and MyHC IIA protein expression in quadriceps femoris (**G**) and gastrocnemius (**H**). Means without a common letter differ, *p* < 0.05. **: *p* < 0.01, ***: *p* < 0.001. Values represent the mean ± SEM.

**Figure 3 ijms-25-06184-f003:**
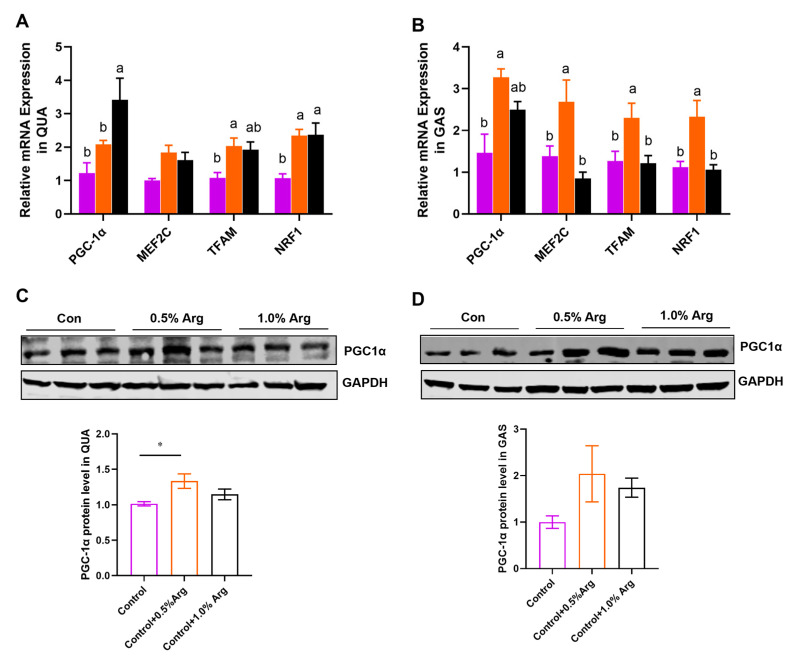
Arg promotes skeletal muscle mitochondrial biosynthesis. (**A**,**B**) The relative mRNA expression of *PGC-1α*, *MEF2C*, *TFAM*, and *NRF1* in quadriceps femoris (**A**) and gastrocnemius (**B**). The purple column represents the control group with no added Arg, the orange column represents the treatment group with 0.5% Arg added. The black column represents the treatment group with 1.0% Arg added. (**C**,**D**) Immunoblots and quantification of PGC-1α in quadriceps femoris (**C**) and gastrocnemius (**D**). Means without a common letter differ, *p* < 0.05. *: *p* < 0.05. Values represent the mean ± SEM.

**Figure 4 ijms-25-06184-f004:**
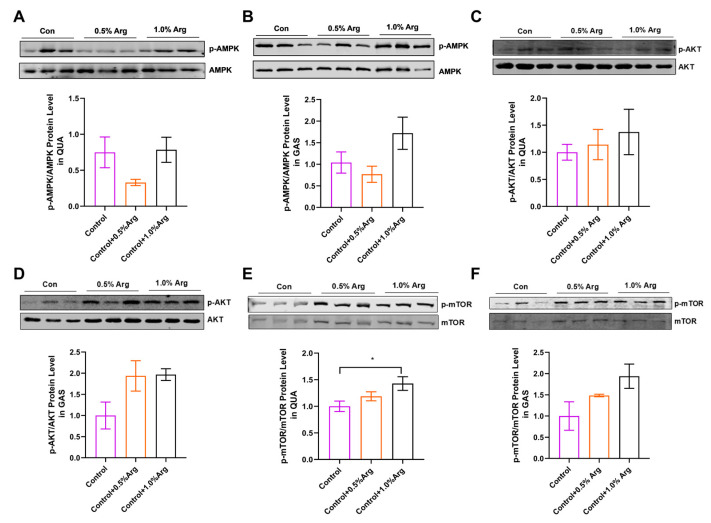
Arg activates mTOR signaling pathway in skeletal muscle of mice. (**A**,**B**) Immunoblots and quantification of p-AMPK protein expression in quadriceps femoris (**A**), and gastrocnemius (**B**). (**C**,**D**) Immunoblots and quantification of p-AKT protein expression in quadriceps femoris (**C**), and gastrocnemius (**D**). (**E**,**F**) Immunoblots and quantification of p-mTOR protein expression in quadriceps femoris (**E**), and gastrocnemius (**F**). *: *p* < 0.05. Values represent the mean ± SEM.

**Figure 5 ijms-25-06184-f005:**
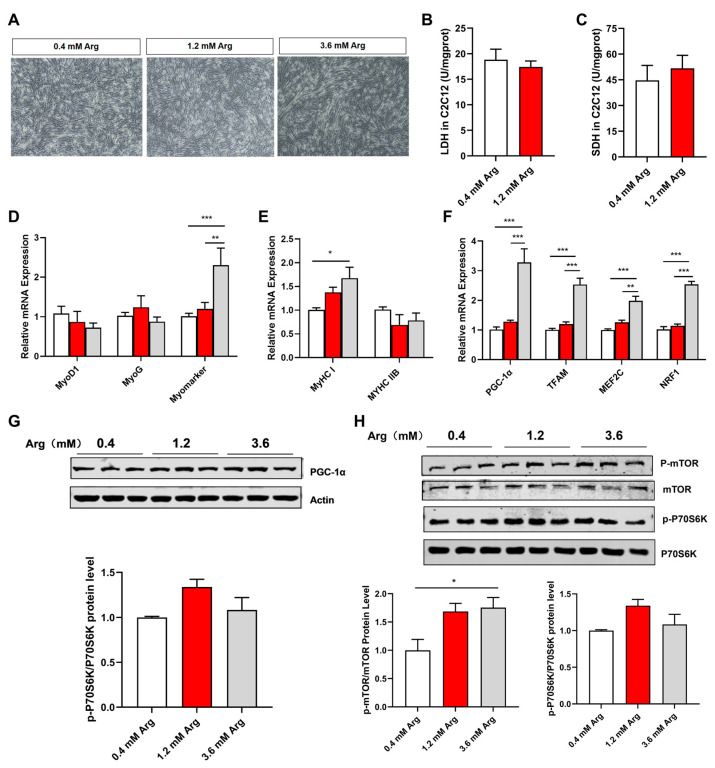
Effects of Arg on expression of MyHC and mTOR signaling pathway in C2C12 cells. Histomorphology of C2C12 cells. Scale bars represent 50 μm (**A**). (**B**,**C**) The enzyme activity of LDH (**B**), and SDH (**C**) in C2C12 cells. (**D**–**F**) The mRNA expression of *MyoD*, *MyoG*, *Myomarker* (**D**), *MyHC I* and *MyHC IIB* (**E**), *PGC-1α*, *TFAM*, *MEF2C*, and *NRF1* (**F**) in C2C12 cells. The white column represents the group with 0.4 mM of Arg added, the red column represents the group with 1.2 mM of Arg added. The gray column represents the group with 3.6 mM of Arg added. (**G**,**H**) The protein expression of PGC-1α (**G**) and mTOR-p70S6K (**H**) in C2C12 cells. *: *p* < 0.05, **: *p* < 0.01, ***: *p* < 0.001. Values represent the mean ± SEM.

**Figure 6 ijms-25-06184-f006:**
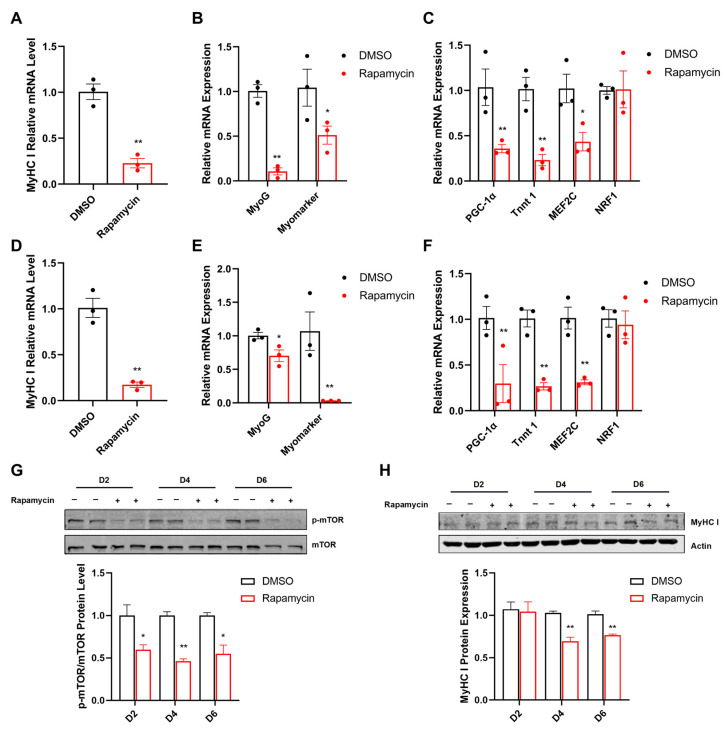
Rapamycin inhibits MyHC I expression and mitochondrial function in C2C12 cells. (**A**–**C**) Relative mRNA expression of *MyHC I* (**A**), *MyoG*, *Myomarker* (**B**), *PGC-1α*, *Tnnt1*, *MEF2C*, and *NRF1* (**C**) on C2C12 cells challenged by rapamycin for 4 days during differentiation. (**D**–**F**) Relative mRNA expression of *MyHC I* (**D**), *MyoG*, *Myomarker* (**E**), *PGC-1α*, *Tnnt1*, *MEF2C*, and *NRF1* (**F**) on C2C12 cells challenged by rapamycin for 6 days during differentiation. (**G**,**H**) The protein expression of mTOR (**G**), and Myhc I (**H**) in C2C12 cells challenged by rapamycin for 2/4/6 days during differentiation. *: *p* < 0.05, **: *p* < 0.01. Values represent the mean ± SEM.

**Figure 7 ijms-25-06184-f007:**
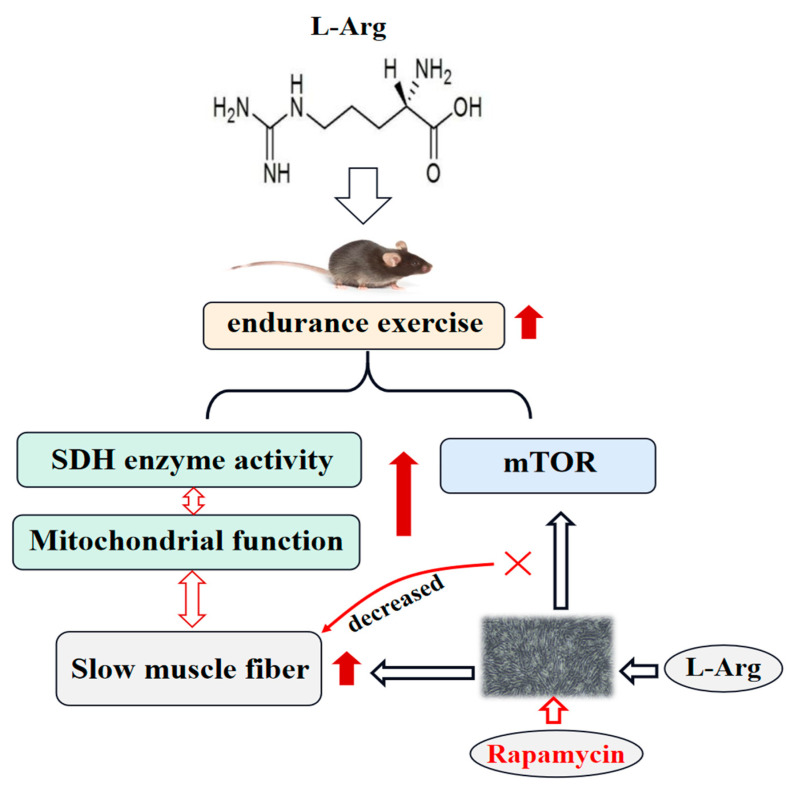
Supplementation with arginine can improve endurance exercise performance, enhance SDH enzyme activity, and upregulate MyHC I/IIA mRNA or protein expression in GAS or QUA, suggesting that arginine induces conversion from fast- to slow-twitch fibers. In addition, arginine promotes mitochondrial biogenesis and activates the Akt/mTOR signaling pathway involved in muscle function and muscle fiber remodeling.

**Table 1 ijms-25-06184-t001:** Primers used in real-time quantitative PCR.

Target Genes	Primer Sequence (5′ to 3′)
*GAPDH*	F: ATGGTGAAGGTCGGAGTGAA R: CGTGGGTGGAATCATACTGG
*MYHC I*	F: GTCAAGGCCAAGATCGTGTC R: CTCCTTCACAGTCACCGTCT
*MYHC IIA*	F: CAGTGTCTAAGGCCAAGGGA R: TCTCATCAAGCTGCCTGGAA
*MYHC IIB*	F: AAGCCTGCCTCCTTCTTCAT R: CAAACACCGATGACTTGGCA
*MYHC IIX*	F: CCAAAGGCAAGGTTGAAGCT R: CAGCCAGCGATGTTGTAGTC
*PGC-1α*	F: GGATATACTTTACGCAGGTCGA R: CGTCTGAGTTGGTATCTAGGTC
*MEF2C*	F: GATCTCCGCGTTCTTATCCC R: CCAATGACTGAGCCGACTG
*TNNT1*	F: TGGATCCACCAGCTGGAATCAGAA R: GCTGATGCGGTTGTAGAGCACATT
*NRF1*	F: GTTGCCCAAGTGAATTACTCTG R: TCGTCTGGATGGTCATTTCAC
*TFAM*	F: GTGAGCAAGTATAAAGAGCAGC R: CTGAACGAGGTCTTTTTGGTTT

## Data Availability

Data will be made available on request.

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
