# Peer review of "Arginine Regulates Skeletal Muscle Fiber Type Formation via mTOR Signaling Pathway"

_ijms, 2024, doi:10.3390/ijms25116184_

Round 1
Reviewer 1 Report
Comments and Suggestions for Authors
The authors studied the effects of arginine on muscle fibre type, showing that arginine modulates skeletal muscle fibre type switch towards slow-twitch fibres and enhances mitochondrial functions by mTOR signalling. The manuscript is interesting, I have a following comments:
1. In introduction the authors could briefly discuss also the effect of fibre type proportion on metabolic health.
2. The quality of Figures is too low and cannot be reviewed since the text in graphs is illegible.
3. The authors found that the proportion of small fibres increased. This is probably due to the fact that in mice slow fibres are usually smaller than fibres type 2b. The proportion and size of each fibre type in each studied muscle should be added to the results.
4. In C57BL/6J mice the gastrocnemius muscle is almost solely composed of fibres type 2b (more than 50%), 2x (around 15%), 2a (around 20 %) and just 1-5% of type 1. Furthermore, in these mice fibres type 2a are more oxidative than type 1 so analysis of MyHC 1 and 2a in figure2 is not very relevant. MyHC 2b and 2x are much more relevant, especially in fast twitch muscles.
5. The limitation is also, that authors probably extracted gastrocnemius muscles together with plantaris muscle, which has higher proportion of fibres type 1 and 2a. This should be mentioned in limitations because it can give significant differences between studying muscle as a whole sample or just studying pure GAS regions of sample.
Author Response
To Reviewer #1
The authors studied the effects of arginine on muscle fiber type, showing that arginine modulates skeletal muscle fiber type switch towards slow-twitch fibers and enhances mitochondrial functions by mTOR signaling. The manuscript is interesting, I have a following comments:
Response: Thank you very much for your efforts on our manuscript. We have addressed each comment point-by-point.
- In introduction the authors could briefly discuss also the effect of fiber type proportion on metabolic health.
Response: In the introduction section, we have added the effect of fiber type proportion on metabolic health: “In humans, slow and fast muscle fibers have a direct impact on endurance and explosive power, respectively. Specifically, increasing the proportion of MyHC I can improve the physical fitness of resistance training athletes, such as weightlifters. Conversely, increasing the proportion of MyHC IIA can enhance the performance of athletes who rely on explosive power, such as sprinters.” at line 43-48.
- The quality of Figures is too low and cannot be reviewed since the text in graphs is illegible.
Response: We have re-adjusted the quality of Figures.
- The authors found that the proportion of small fibres increased. This is probably due to the fact that in mice slow fibres are usually smaller than fibres type 2b. The proportion and size of each fibre type in each studied muscle should be added to the results.
Response: During the experiment, we also took into account the influence of the size of slow muscle and type 2 muscle fibers on the result of the transformation of muscle fiber type by arginine. Therefore, in the follow-up result 2.3, we further analyzed the influence of arginine on the transformation of each type of muscle fiber, and the results showed that arginine increased the mRNA expression level of MyHC I. Meanwhile, the mRNA expression level of MyHC IIB was decreased. These results determined which small fibers were increased by arginine at the molecular level.
We have supplemented the diagram notes (the description of muscle fiber type in the immunofluorescence staining) in line 142-144 and modified the corresponding results in line 129-131.
- In C57BL/6J mice the gastrocnemius muscle is almost solely composed of fibres type 2b (more than 50%), 2x (around 15%), 2a (around 20 %) and just 1-5% of type 1. Furthermore, in these mice fibres type 2a are more oxidative than type 1 so analysis of MyHC 1 and 2a in figure2 is not very relevant. MyHC 2b and 2x are much more relevant, especially in fast twitch muscles.
Response: We did not clearly explain the reason for our analysis of MyHC I and MyHC IIA in the results in the previous version, thus causing confusion. We would like to explain in the revised version that in result 2.2 section, we found that arginine can improve the muscle endurance of mice. Therefore, In Result 2.3, we further explored the effect of arginine on the muscle fiber transformation of MyHC I and MyHC IIA, which are related to muscle endurance. Through the test, it was found that the addition of arginine promoted the increase of the proportion of these two muscle fibers, and concluded that arginine could increase the proportion of slow muscle fibers and improve muscle endurance. We have revised the description of result 2.3 in lines 123-140.
- The limitation is also, that authors probably extracted gastrocnemius muscles together with plantaris muscle, which has higher proportion of fibres type 1 and 2a. This should be mentioned in limitations because it can give significant differences between studying muscle as a whole sample or just studying pure GAS regions of sample.
Response: Since the plantaris muscle and the gastrocnemius muscle of mice are small in size and close in distribution, it is true that the plantaris muscle may also be collected when the sample of the gastrocnemius muscle is collected. We have added a discussion on this limitation in line 258-265 of the discussion section. The revised version is as follows:“ In addition, it is important to acknowledge certain limitations in investigating the effects of arginine on different muscle fiber types. Specifically, due to the small size and close proximity of various types of skeletal muscles in mice, there is a possibility of inadvertently collecting nearby muscles during sampling, which could potentially influence the analysis of the collected samples. For instance, the gastrocnemius and plantaris muscles have a close distribution, and it is possible that some plantaris muscle samples were inadvertently collected along with the gastrocnemius samples, potentially affecting the obtained results.”
Reviewer 2 Report
Comments and Suggestions for Authors
The article titled "Arginine Regulates Skeletal Muscle Fiber Type Formation via 2 mTOR signaling pathway" by Min Zhou et al. is an interesting study that explores the role of Arg in skeletal muscle fiber composition and mitochondrial function through the mTOR signaling pathway. The study is well-planned, experiments were conducted appropriately, the results were presented rationally, and the discussion was done in the available light of literature. The authors concluded that Arg modulates skeletal muscle fiber type towards slow-twitch fibers and enhances mitochondrial functions by upregulating gene expression through the mTOR signaling pathway. However, the following comments should be addressed. The manuscript can be considered for publication after addressing these comments.
Comments:
1. How did the authors determine the Arginine (0.5% and 1.0 % arginine) dose in animal studies?
2. All the figures are blurred; replace them with better quality. It is very hard to comprehend the figure.
3. Rewrite the result for more clarity. What do you mean by the above genes in the given statement? “In mixed GAS, exogenous addition of 0.5% Arg up-regulated mRNA level of above genes, and enhance PGC-1α protein level without statistical significance which may be related to the difference within the group (Figure 3B, D).”
4. Figure 4: In the result section for Figures 4A and 4B, the authors should clarify whether they are referring to the total AMPK level in the tissue compared to its phosphorylated protein or use p-AMPK levels. The same clarification is needed for the figures CD. Additionally, it would be helpful if authors split the AKT and MTOR blots with respective graphical representations.
5. Why did the authors treat C2C12 cells with 0.4, 1.2, and 3.6 mM Arg? Is there any physiological relevance? The treatments were not described in the methods. Why?
6. Figure 5: there is a discrepancy between the results written and the graphs. It should be rewritten carefully with more clarity, especially D-F. Some parameters had statistical significance, while others did not.
7. What happens to Arginine levels in obesity? Does Arg have any effect on skeletal muscle fiber in obesity?
Comments on the Quality of English LanguageThe introduction should be rewritten for more clarity, e.g., “It not only influences human exercise ability and health [2] but also directly impacts the yield and quality of meat in livestock and poultry”. Change influence to affect for clarity.
Results for Figures 4 and 5 should be rewritten for more clarity.
Author Response
To Reviewer #2
The article titled "Arginine Regulates Skeletal Muscle Fiber Type Formation via 2 mTOR signaling pathway" by Min Zhou et al. is an interesting study that explores the role of Arg in skeletal muscle fiber composition and mitochondrial function through the mTOR signaling pathway. The study is well-planned, experiments were conducted appropriately, the results were presented rationally, and the discussion was done in the available light of literature. The authors concluded that Arg modulates skeletal muscle fiber type towards slow-twitch fibers and enhances mitochondrial functions by upregulating gene expression through the mTOR signaling pathway. However, the following comments should be addressed. The manuscript can be considered for publication after addressing these comments.
Response: Thank you very much for your efforts on our manuscript. We have addressed each comment point-by-point.
- How did the authors determine the Arginine (0.5% and 1.0 % arginine) dose in animal studies?
Response: Before using arginine to treat mice, we found many studies showed that 0.5% and 1.0% arginine had an effect on skeletal muscle of animals[1, 2]. Therefore, we try to use the dosage of arginine used in other studies to treat mice, and the results showed that 0.5% and 1.0% level of arginine did have an effect on mice. The revised version of the manuscript includes references to support the chosen arginine dosage.
- All the figures are blurred; replace them with better quality. It is very hard to comprehend the figure.
Response: We have re-adjusted the quality of Figures.
- Rewrite the result for more clarity. What do you mean by the above genes in the given statement? “In mixed GAS, exogenous addition of 0.5% Arg up-regulated mRNA level of above genes, and enhance PGC-1α protein level without statistical significance which may be related to the difference within the group (Figure 3B, D).”
Response: The term "above genes" refers to PGC-1α, MEF2C, TFAM, and NRF1, as mentioned in line 154. The revised version is as follows: In the mixed GAS, the addition of 0.5% arginine from an external source resulted in the up- regulation of mRNA levels for PGC-1α, MEF2C, TFAM, and NRF1 (Figure 3B). While there was an observed enhancement in PGC-1α protein levels, it did not reach statistical significance, possibly due to variations within the group (Figure 3D).
- Figure 4: In the result section for Figures 4A and 4B, the authors should clarify whether they are referring to the total AMPK level in the tissue compared to its phosphorylated protein or use p-AMPK levels. The same clarification is needed for the figures CD. Additionally, it would be helpful if authors split the AKT and MTOR blots with respective graphical representations.
Response: We have clarified that we are referring to the p-AMPK levels in Figure 4A and 4B, also revised the description of Figure 4C-4F at line 172-177. Moreover, we have split the AKT and mTOR blots with respective graphical representations. Here is the new Figure 4:
Figure 4. Arg activates mTOR signaling pathway in skeletal muscle of mice.
- Why did the authors treat C2C12 cells with 0.4, 1.2, and 3.6 mM Arg? Is there any physiological relevance? The treatments were not described in the methods. Why?
Response: The concentration of 0.4mM arginine in the control group was the concentration of arginine contained in the growth medium (high glucose DMEM medium) without additional arginine. The concentration of 1.2mM and 3.6mM arginine in the treatment group was a reference to the dosage of arginine treated with C2C12 cell line in other researches[1, 3, 4], and there was no physiological correlation. We apologize for missing the description of arginine treatment of C2C12 cells in the method, which has been supplemented in line 187-191 of the result section and in line 364-367 of the methods section.
- Figure 5: there is a discrepancy between the results written and the graphs. It should be rewritten carefully with more clarity, especially D-F. Some parameters had statistical significance, while others did not.
Response: We have rewritten the results of Figure 5 in lines 191-198 to provide greater clarity.
- What happens to Arginine levels in obesity? Does Arg have any effect on skeletal muscle fiber in obesity?
Response: Most studies indicate reduced arginine levels in obesity. For example, Tatsuo et al. found significantly decreased plasma arginine levels in a high fat diet-induced obesity mouse model [5]. Raven et al. found that while the concentration of arginine remained relatively stable in obese individuals, the production of de novo arginine and the overall production of arginine-related amino acids were reduced in obese women [6]. These findings suggest altered arginine metabolism in obesity, with lower arginine levels being negatively correlated with the development of obesity. It is possible that the demand for arginine and its related amino acids increase s in obesity.
Furthermore, arginine exerts effects on skeletal muscle in obesity. Studies have found that oral supplementation of arginine in diet-induced obese rats increases the weight of soles muscle and extensor digitorum longus muscle, promotes the oxidation of energy substrates in skeletal muscle, and thus reduces white adipose tissue in the body[7, 8].
- The introduction should be rewritten for more clarity, e.g., “It not only influences human exercise ability and health [2] but also directly impacts the yield and quality of meat in livestock and poultry”. Change influence to affect for clarity.
Response: We have replaced "influence" with "affect" in our manuscript. Additionally, we conducted a thorough review of the introduction and all other sections to ensure clear and accurate expression throughout.
- Results for Figures 4 and 5 should be rewritten for more clarity.
Response: We have revised the descriptions for the results of Figures 4 and 5 to improve clarity and accuracy.
References
- Chen, X.; Guo, Y.; Jia, G.; Liu, G.; Zhao, H.; Huang, Z. Arginine promotes skeletal muscle fiber type transformation from fast-twitch to slow-twitch via Sirt1/AMPK pathway. J Nutr Biochem. 2018, 61, 155-162.
- Chen, X.; Luo, X.; Chen, D.; Yu, B.; He, J.; Huang, Z. Arginine promotes porcine type I muscle fibres formation through improvement of mitochondrial biogenesis. Br J Nutr. 2020, 123, 499-507.
- Gong, L.; Zhang, X.; Qiu, K.; He, L.; Wang, Y.; Yin, J. Arginine promotes myogenic differentiation and myotube formation through the elevation of cytoplasmic calcium concentration. Anim Nutr. 2021, 7, 1115-1123.
- Zhao, Y.; Jiang, Q.; Zhang, X.; Zhu, X.; Dong, X.; Shen, L.; Zhang, S.; Niu, L.; Chen, L.; Zhang, M.; Jiang, J.; Chen, D.; Zhu, L. l-Arginine Alleviates LPS-Induced Oxidative Stress and Apoptosis via Activating SIRT1-AKT-Nrf2 and SIRT1-FOXO3a Signaling Pathways in C2C12 Myotube Cells. Antioxidants (Basel). 2021, 10.
- Ito, T.; Kubo, M.; Nagaoka, K.; Funakubo, N.; Setiawan, H.; Takemoto, K.; Eguchi, E.; Fujikura, Y.; Ogino, K. Early obesity leads to increases in hepatic arginase I and related systemic changes in nitric oxide and L-arginine metabolism in mice. J Physiol Biochem. 2018, 74, 9-16.
- Wierzchowska-McNew, R. A.; Engelen, M.; Thaden, J. J.; Ten Have, G. A. M.; Deutz, N. E. P. Obesity- and sex-related metabolism of arginine and nitric oxide in adults. Am J Clin Nutr. 2022, 116, 1610-1620.
- Jobgen, W. S.; Lee, M. J.; Fried, S. K.; Wu, G. l-Arginine supplementation regulates energy-substrate metabolism in skeletal muscle and adipose tissue of diet-induced obese rats. Exp Biol Med (Maywood). 2023, 248, 209-216.
- Jobgen, W.; Meininger, C. J.; Jobgen, S. C.; Li, P.; Lee, M. J.; Smith, S. B.; Spencer, T. E.; Fried, S. K.; Wu, G. Dietary L-arginine supplementation reduces white fat gain and enhances skeletal muscle and brown fat masses in diet-induced obese rats. J Nutr. 2009, 139, 230-7.

Round 2
Reviewer 1 Report
Comments and Suggestions for Authors
The authors satisfactorily addressed my comments.